# Existential aspects of fatherhood transition— A systematic qualitative review protocol using framework synthesis

Marie Sejrsgaard[1,2]*, Caroline Moos[2,3], Maiken Fabricius Damm[1],
Christina Prinds[1,2,4]

1 Department of Womens Health, University Hospital of Southern Denmark, Aabenraa, Denmark,
2 Department of Regional Health Research, University of Southern Denmark, Odense, Denmark,
3 Department of Clinical Research, University Hospital of Southern Denmark, Aabenraa, Denmark,
4 Research Unit of Gynecology and Obstetrics, Odense University Hospital, Odense, Denmark,

☻ These authors contributed equally to this work.
* Guldmarie@gmail.com

## Abstract

### Background

Research on the experiences of parenthood transition has previously primarily focused on the mother/birthing woman. Even with an expanding interest in the existential aspects of health in relation to becoming a parent, the experiences of the fathers are often overlooked, as in a 2014 scoping review exploring the existential meaning-making of becoming a mother. With changing expectations of the role of fatherhood, a greater understanding of the experiences of fathers is required, and a specific focus on the existential aspects could add valuable insight into changes in identity, values, and notions of a good life among new and expecting fathers.

### Methods and analysis

This systematic qualitative review protocol maps out a plan to explore existing knowledge on the existential aspects of becoming a father through a Framework Synthesis using Peter la Cours model of existential health. Five databases (MEDLINE, Embase, PsycINFO, CINAHL and Web of Science) will be searched from 01.01.2010 to the present day. Two independent, blinded reviewers will assess the eligibility of included studies according to the listed inclusion and exclusion criteria. The Critical Appraisal Skills Program (CASP) checklist will be utilized to assess the methodological quality of the included studies. Additionally, Web of Science will be used for snowball searching of included references.

**Data availability statement:** No datasets were generated or analysed during the current study. All relevant data from this study will be made available upon study completion.

**Funding:** The author(s) received no specific funding for this work.

**Competing interests:** The authors have declared that no competing interests exist.

## Conclusion and dissemination

The findings from this synthesis may contribute to development of evidence-informed clinical guidelines that promote person-centred and father-inclusive maternal care. The findings will be disseminated in peer-reviewed journals, conferences, and social media.

## Registration

The study protocol has been registered with the International Platform for Prospective Registration of Systematic Reviews (PROSPERO), with registration number: CRD420251038896.

## Introduction

Although there has been a considerable amount of research on the transition to parenthood, much of it has focused on becoming a mother [1]. Research on fatherhood is of relatively newer origin and has shifted focus according to the concurrent cultural and sociological expectations of the fatherhood role. The sociological research that primarily focused on the social and cultural importance of the presence of a father figure in the home, often utilizing a definition of the father as an emotionally distant "bread-winner" has evolved to psychological explorations of the fathers as equally capable of creating and maintaining healthy attachment patterns in the child and newborn [2,3]. These changes have occurred over a relatively short period of time, potentially leaving new fathers without the kind of role models they need to navigate fatherhood in the present day [4]. Over the last few decades there has been an increase in the expectations of fathers involvement and participation in pregnancy and childbirth [5]. Nonetheless some fathers still feel that they are viewed and treated in a maternity care system as something between a visitor and a support-person for the birthing woman [6]. It could be argued that the underlying interest in the well-being of the father is an exploration of the positive effect his well-being can have on the well-being of the mother and child [7]. This is also reflected in national guidelines for maternity care such as those in Denmark, where the focus is more on preparation, motivation and action during childbirth, and later support for the mother-child dyad, with less emphasis on supporting the fathers individual experiences and integration of the fatherhood transition [8]. The same dynamic is reflected in the NICE guidelines from the United Kingdom [9], or in the WHO recommendations on antenatal care for a positive birth experience [10].

A literature review on men's experiences of fatherhood transition was conducted in 2009 [4], with the main focus being on the psychological, interrelational and social dimensions of fatherhood. A fourth dimension of health has been introduced by WHO as an integral part of accessing public health, and is defined as spiritual or existential health, where a person is: *"…able to deal with day-to-day life in a manner which leads to the realization of one's full potential; meaning and purpose*

*of life; and happiness from within*"[11]. In a well-cited scoping review from 2014, the existential dimensions of a healthy meaning-making of the transition to motherhood were explored through the theory of existential psychologist Emmy van Deurzen [12]. No similar review from the perspective of fathers has yet been found, despite recent research suggesting that fatherhood affects men existentially: identity, values, and notions of a good life are changed and challenged [4,13]. From an existential philosophical perspective, considerations about responsibility, career, and masculinity emerge, influencing the experience of becoming a father [14–16]. In this light, the perspective of existential health seems relevant to explore in the transition to fatherhood, since it could give a more direct input to healthcare professionals on how to better support the fathers as individuals with their own needs, and not merely as next-of-kin to the birthing woman.

The proposed systematic review will be conducted parallel with a systematic review on existential meaning-making for mothers, and both will synthesise the latest literature from the past 15 years, utilizing the newest model of Existential Health by Peter la Cour [17]. The aim of the systematic review is to synthesise experiences related to existential meaning-making during the transition to fatherhood.

## Materials and methods

The formation of this review protocol has been guided by Butler et al (2016) [18], and is reported according to the PRISMA-P (Preferred Reporting Items for Systematic Reviews and Meta-Analysis Protocols) for transparency [19]. The systematic review will be conducted in six phases inspired by the Cochrane guide for performing systematic qualitative reviews [20] as outlined below. The results will be reported according to ENTREQ (Enhancing Transparency in Reporting the Synthesis of Qualitative Research) guidelines [21] to ensure the methodological rigour of conducting and reporting the review.

### Phase 1: Defining the review scope and formulating the review question

The research question is based on the PICo (Population, Interest, Context) typology [18]. This typology helps define the scope of the research as well as outline the inclusion and exclusion criteria [18], and has been shown to have the highest sensitivity to identifying existing knowledge [22]. The research question is: "How do fathers make existential meaning in the transition to fatherhood?"

### Phase 2: Searching for and identifying relevant studies

Five databases (Embase (OVID), MEDLINE (OVID), PsycINFO (OVID), CINAHL (EBSCO) and Web of Science) will be searched from 01.01.2010 until the present day. After the search, screening and identification of relevant studies, the Web of Science database will be used for forward citation to identify any additional studies. The search will consist of three blocks where the population-block is constructed using search terms pertaining to parenthood, fatherhood or motherhood, as well as related synonyms. The Interest-block focuses on the transitional aspect of becoming a parent and includes the search terms birth, labour, transition and related synonyms. The context-block will consist of search terms related to existentialism, life change events and meaning-making thus narrowing the scope of results to focus the search on articles relating to the existential aspects of the parenthood transition. Each block will be constructed using relevant medical subject headings according to the chosen database as well as free text words combined with the Boolean operator OR. These blocks will be combined with the Boolean operator AND [23,24]. A fourth block will be added, to focus the search on the qualitative methodologies, since it is construed that we can only expect the research question to be explored through these methods. A preliminary search string was performed in Embase on April 23rd, 2025, to validate the relevance of the constructed blocks. See Supporting Information section, S1 File.

Grey literature searching: ProQuest Dissertations & Theses will be systematically searched with relevant subject headings and filters related to the review topic. In Google Scholar the first 200 results will be sorted by relevance and screened

using a simplified version of the above-mentioned search strategy. The search string will initially be performed as one string for both reviews on fatherhood and motherhood, as there is likely a considerable overlap of articles addressing mothers' and fathers' perspectives respectively. The selection of relevant articles for each review will be made during the selection process, according to inclusion and exclusion criteria.

### Phase 3: Defining inclusion and exclusion criteria

Qualitative studies published between 2010 and 2025 will be included in the review, provided they are conducted within a Western context. Western contexts are defined as regions that encourage individuals to express their personal goals and identities. These regions share common characteristics while also embracing a high degree of cultural diversity and inclusion. Countries included in this definition are European Nations, Canada, North America, Australia, and New Zealand. The language of publication will not serve as a restriction for the inclusion of studies.

The included studies will be original studies from peer-reviewed journals focusing on men's descriptions of the experience of transitioning to fatherhood. All types of fatherhood will be included, thus also considering adoption, fatherhood achieved through surrogacy pregnancy or rainbow family structures, if there is specific qualitative data on the experiences of fatherhood transition. Studies including other perspectives will be included if the fathers' experiences are reported explicitly. Studies focusing on pregnancy loss or solely on postpartum depression will be excluded.

### Phase 4: Structuring and reviewing the search results

All relevant studies will be uploaded to Covidence, an electronic tool for conducting reviews and the PRISMA 2020 flow-chart will document the results of the screening process. Title/Abstract screening and full text screening will be conducted by two blinded reviewers (MS and CM). Studies not qualifying for inclusion at full-text screening will be excluded with justification. CP (the first author of the original scoping review on motherhood) and MFD will adjudicate any discrepancies.

### Phase 5: Critical appraisal of included studies

The CASP (Critical Appraisal Skills Programme) will assess the individual studies' methodological rigour, credibility and relevance [25]. The quality of these studies will be appraised collaboratively with at least two researchers. While studies of lower quality will not be excluded from the review, their limitations will be considered during the synthesis and discussion phases to provide a balanced interpretation of the findings.

### Phase 6: Data extraction and methodology of synthesis

Covidence software will be used to extract study characteristics, including author, name of journal, year of publication, study location, study aims, participant characteristics, data collection methods, parity, spiritual association, timing and duration of interviews.

Nvivo software will be used to extract first-order constructs (direct citations from participants) or second-order constructs (other citations and interpretations) as both have been shown to support data extraction of interest for qualitative reviews [26]. The chosen methodology is Framework Synthesis, as it offers consistency and transparency in coding and adds structure to the synthesis process [27]. The chosen a priori framework is the theory of Peter la Cour, who in *Introducing Existential Health (2025)* [17] renames and redefines the four dimensions of health suggested by Emmy van Deurzen (2005) [28–30], and subsequently adds more detail to the existential health dimension, through an exploration of four layered components of it. La Cours theory will be used in an iterative and dynamic approach to interpret the data through different expressions of existential health according to the specific existential components. The stages of interpretation will be 1) familiarizing the data in an inductive approach and development of initial existential codebook. 2) indexing the data into the a priori framework in iterative cycles of deductive and augmentative approaches, adjusting the framework and codes to fit the data. 3) Emerging patterns, relationships and themes will be elaborated and refined in depth through

discussions between the first author and coauthors to ensure accuracy and consistency in capturing the depth of the individual expressions of existential health. To support the rigour of the framework synthesis, an audit trail will be kept to document how coding decisions were made, how data were charted onto the preliminary framework, and how themes or relationships were adapted throughout the synthesis.

### Ethics statement

As a systematic qualitative review is secondary research based on publicly available materials, ethical approval is not required.

## Discussion

Over the last decade, there has been an increasing debate about the lack of focus on the experiences and health issues of new fathers [31]. Publications have been written by the Danish NGO Forum for Men's Health, attempting to address this issue and help qualify how healthcare professionals can adopt a more father-inclusive mode of conduct in their practice [32–34].

By synthesizing the existing literature, this review aims to contribute to reflections among healthcare professionals in the maternity care system, healthcare leaders and policymakers on the often-unspoken needs of fathers-to-be. The findings will contribute to a deeper understanding of previously unexplored aspects of health, thus adding nuances and improving the quality of fathers' experiences within the maternity care system.

### Strengths and limitations

A strength of this protocol is that it proposes an expansion of a well-cited scoping review on how women make existential meaning during the transition to motherhood first published in 2014, thus expanding the understanding of existential aspects and considerations in parenthood transitions to include reflections from the fathers. This focus on fatherhood is the first of its kind, recognizing that although there may be overlaps in the experience of existential meaning-making during the parenthood transition for mothers and fathers, a sole focus on fathers allows unique themes related to, e.g., gender roles and expectations to emerge. The review adheres to international reporting guidelines, and utilizes stringent, transparent, and reproducible methods to ensure methodological rigor in the screening, data extraction and analysis phases with two independent reviewers.

Some limitations must also be addressed; firstly, that the strict focus on qualitative data may lead to selection bias. Secondly, the concept of 'existential aspects' is difficult to construct as it is not easily operationalized into a search block, which introduces the risk that the sensitivity of the search omits important studies. Thirdly, while biological fatherhood is universal, in the context of this review, the notion of fatherhood is also a social construct, and as such, the results are culturally dependent. This could, in turn, challenge the transferability and applicability of the results to other cultural contexts.

## Supporting information

**S1 File. Example of search string, EMBASE(OVID).**
(DOCX)

**S2 File. PRISMA-P 2015 checklist.**
(DOCX)

## Author contributions

**Conceptualization:** Marie Sejrsgaard, Caroline Moos.

**Methodology:** Marie Sejrsgaard, Caroline Moos.

**Project administration:** Christina Prinds.

**Supervision:** Christina Prinds.

**Writing – original draft:** Marie Sejrsgaard.

**Writing – review & editing:** Caroline Moos, Maiken Fabricius Damm, Christina Prinds.

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
