## [Editor Report · Decision Letter 0]

20 Oct 2025

Dear Dr. Sejrsgaard,

Thank you for submitting your manuscript to PLOS ONE. After careful consideration, we feel that it has merit but does not fully meet PLOS ONE’s publication criteria as it currently stands. Therefore, we invite you to submit a revised version of the manuscript that addresses the points raised during the review process.

We look forward to receiving your revised manuscript.

Kind regards,

Sergi Fàbregues

Academic Editor

PLOS ONE

Additional Editor Comments:

This is a well-reported systematic review protocol on an interesting and well-justified topic. Please find my recommendations below for improving the manuscript:

- Please clarify whether the protocol has been registered.

- Please provide a rationale for choosing the four selected databases. Given the multidisciplinary nature of the topic, it is surprising that broad, multidisciplinary databases such as Web of Science and Scopus will not be used.

- Please provide more details on how the framework synthesis approach will be applied, including the steps to be implemented.

- The discussion section is missing. A complete discussion section needs to be included.

- No limitations are mentioned in the Strengths and Limitations section. Please address this.

- An explanation of how the strength of the body of evidence will be assessed is absent and should be provided.

- Several typos are present throughout the manuscript and should be corrected.
---

## [Author Response · Author response to Decision Letter 1]

14 Nov 2025

The required changes have been made to the formatting of the manuscript and the file naming.

The ethics statement has been removed from the abstract and is now only to be found as a Subheading under the Method section (see line 144-147).

There was no such recommendation.

Thank you, the reference list has been reviewed, and no changes have been made apart from adding four new references corresponding to the added discussion section.

Additional Editor Comments:

This is a well-reported systematic review protocol on an interesting and well-justified topic. Please find my recommendations below for improving the manuscript:

Thank you for this comment.

- Please clarify whether the protocol has been registered.

The protocol is registered in PROSPERO (see line 21-22).

- Please provide a rationale for choosing the four selected databases. Given the multidisciplinary nature of the topic, it is surprising that broad, multidisciplinary databases such as Web of Science and Scopus will not be used.

Thank you for this valuable feedback. After discussion and on your recommendation, we decided to include Web of Science on the list of databases to be searched.

- Please provide more details on how the framework synthesis approach will be applied, including the steps to be implemented.

Thank you for this opportunity. Please see the addition in line 127-139 to address this comment.

- The discussion section is missing. A complete discussion section needs to be included.

Some protocol articles include a discussion or elaboration on the introduction in the discussion. As the review is not completed it is subject to debate whether a discussion section is of value. We have chosen to expand on some of our reflections on the relevance and implications on the topic of this review in the discussion section (see line 148-157).

- No limitations are mentioned in the Strengths and Limitations section. Please address this.

Thank you for drawing this omission to our attention. The Strengths and Limitations section has been rewritten for a more coherent style, and to include some of the challenges of a review of this nature (see line 158-173).

- An explanation of how the strength of the body of evidence will be assessed is absent and should be provided.

Thank you for this comment. As our study is qualitative rather than quantitative, the concept of “strength of evidence” is not directly applicable. Instead, we assess the trustworthiness of the evidence, which is evaluated based on methodological rigor rather than statistical strength [1-3]. To ensure transparency and consistency, we will assess selectiveness of reporting by using the established appraisal tool; Critical Appraisal Skills Programme (CASP) checklist (see line 115-118). Additionally, we have added a level of meta reflection to the analysis process to enhance trustworthiness of the results (see line 139-142).

- Several typos are present throughout the manuscript and should be corrected.

We have submitted the manuscript to a third round of proofreading by a native English speaker.

1. Malterud, K., V.D. Siersma, and A.D. Guassora, Sample size in qualitative interview studies: guided by information power. Qualitative health research, 2016. 26(13): p. 1753–1760.

2. Williams, V., A.-M. Boylan, and D. Nunan, Qualitative research as evidence: expanding the paradigm for evidence-based healthcare. BMJ evidence-based medicine, 2019. 24(5): p. 168–169.

3. Williams, V., et al., Appraising qualitative health research-towards a differentiated approach. BMJ Evid Based Med, 2022. 27(4): p. 212–214.

---

## [Editor Report · Decision Letter 1]

17 Nov 2025

Existential aspects of fatherhood transition. -A systematic qualitative review protocol using framework synthesis

PONE-D-25-52335R1

Dear Dr. Sejrsgaard,

We’re pleased to inform you that your manuscript has been judged scientifically suitable for publication and will be formally accepted for publication once it meets all outstanding technical requirements.

Kind regards,

Sergi Fàbregues

Academic Editor

PLOS ONE

Additional Editor Comments:

Just one minor comment: The strength or confidence of evidence in qualitative synthesis is typically assessed using the GRADE-CERQual tool. However, there is no need to modify this in the manuscript; the authors’ response to my comment is appropriate.

---

## [Editor Report · Acceptance letter]

PONE-D-25-52335R1

PLOS ONE

Dear Dr. Sejrsgaard,

I'm pleased to inform you that your manuscript has been deemed suitable for publication in PLOS ONE. Congratulations! Your manuscript is now being handed over to our production team.

Kind regards,

on behalf of

Dr. Sergi Fàbregues

Academic Editor

PLOS ONE